# DNA Mutagenicity of Hydroxyhydroquinone in Roasted Coffee Products and Its Suppression by Chlorogenic Acid, a Coffee Polyphenol, in Oxidative-Damage-Sensitive SAMP8 Mice

**DOI:** 10.3390/ijms25020720

**Published:** 2024-01-05

**Authors:** Keiko Unno, Kyoko Taguchi, Tadashi Hase, Shinichi Meguro, Yoriyuki Nakamura

**Affiliations:** 1Tea Science Center, University of Shizuoka, 52-1 Yada, Suruga-ku, Shizuoka 422-8526, Japan; gp1719@u-shizuoka-ken.ac.jp (K.T.); yori.naka222@u-shizuoka-ken.ac.jp (Y.N.); 2Research and Development, Kao Corporation, 2-1-3 Bunka, Sumida-ku, Tokyo 131-8501, Japan; hase.tadashi@kao.com; 3Biological Science Research, Kao Corporation, Akabane, Ichikai-machi, Haga-gun 321-3497, Japan; meguro.shinichi@kao.com

**Keywords:** chlorogenic acid, coffee polyphenol, glucosuria, hydroxhydroquinone, hydrogen peroxide, SAMP8, oxidative damage, 8-oxoguanine DNA glycosylase 1 (*Ogg1*), polyuria

## Abstract

Hydroxyhydroquinone (HHQ) is an oxidative component produced by roasting coffee beans and has been reported to generate relatively large amounts of reactive oxygen species (ROS). In this study, we used senescence-accelerated mouse prone 8 (SAMP8) mice to determine whether HHQ consumption increases oxidative-stress-induced injury, because in SAMP8 mice, the activity of 8-oxoguanine DNA glycosylase 1, which repairs oxidative modifications in DNA, is decreased. The results showed that two out of twelve (16.7%) HHQ-treated mice presented polyuria and glucosuria around 2 months after the start of treatment, indicating that HHQ may act as a mutagen against SAMP8 mice, which is sensitive to oxidative damage. No abnormalities were observed in the chlorogenic acid (coffee polyphenol, CPP)-treated group. The concentration of hydrogen peroxide in the serum of SAMP8 mice was significantly higher than that in SAMR1 (senescence-resistant) control mice, and the concentration was further increased in the HHQ-treated group. CPP, when coexisting with HHQ at the rate contained in roasted coffee, decreased the amount of hydrogen peroxide in the serum of SAMP8 mice. Although CPP can act both oxidatively and antioxidatively as a polyphenol, CPP acts more antioxidatively when coexisting with HHQ. Thus, the oxidative effect of HHQ was shown to be counteracted by CPP.

## 1. Introduction

Coffee contains coffee polyphenols (CPPs) consisting mainly of caffeoyl quinic acid, feruloyl quinic acid, and dicaffeoyl quinic acid. Chlorogenic acid is the generic name for the ester compounds of these cinnamic acid derivatives and quinic acid [1]. CPP is expected to have antioxidant effects. When combined with a milk fat globule membrane, which was reported to improve motor function, CPP inhibited the age-related decline in brain function in an accelerated aging model mouse (SAMP8) and reduced mouse mortality [2]. This result indicates that the continuous intake of both compounds has a preventive effect on aging. On the other hand, hydroxyhydroquinone (HHQ), an oxidizing component produced by roasting coffee beans, generates relatively large amounts of reactive oxygen species and is thought to act as an oxidative stress loader to the organism [3,4]. Indeed, hydrogen peroxide has been detected in urine in a single human intake evaluation of regular coffee containing HHQ produced by roasting [5,6]. Such reactive oxygen species are thought to reflect oxidative stress in the body and have been shown, for example, to attenuate the hypotensive effects of CPP [7].

SAMP8 mice are characterized by accelerated aging and a shorter lifespan, as well as reduced brain function from a relatively early age [8]. In SAMP8 mice, relevant activities are performed by 8-oxoguanine DNA glycosylase 1 (*Ogg1*), which repairs oxidative modifications in DNA and was reported to be lower than the activities in SAMR1 (senescence-resistant) mice, which have a very similar genetic background to SAMP8 but show normal aging [9]. The mutation in *Ogg1* was found to be a common feature of SAMP-lineage mice with accelerated senescence [10]. Thus, this mutation may contribute to susceptibility to disease via defects in DNA repair including accelerated senescence and a shortened lifespan.

It was reported that the oral administration of HHQ to rats enhanced lipid peroxidation in the lungs and heart but did not enhance DNA damage as indexed by 8-hydroxy-2′-deoxyguanosine [11]. However, we hypothesized that the effects of HHQ may be detectable in SAMP8 mice, which are more sensitive to oxidative injury. In this study, we examined the effect of HHQ on SAMP8 and the involvement of CPP. Then, we explored the effects of HHQ and CPP ingestion on *Ogg1* expression. In addition, since CPP was reported to protect against oxidative damage via phosphatidylinositol-3 kinase (PI3K)/serine-threonine protein kinase (Akt)-mediated activation of the nuclear factor-erythroid 2-related factor 2 (Nrf2)/heme oxygenase (HO-1) pathway [12], the expression of these mRNAs was examined. Nrf2 plays a critical role in controlling the expression of antioxidant genes [13]. The ratio of CPP to HHQ was based on the ratio actually present in coffee.

## 2. Results

### 2.1. Effects of HHQ and CPP Intake on Body Weight and Glucosuria

SAMP8 mice two months of age were fed diets containing 0.02% HHQ and 2% CPP for one month or eight months, and tissues from the livers and brains of young (3-month-old, 3 M) and old (10-month-old, 10 M) mice were collected (Figure 1). Kidneys were collected to obtain samples from only old mice. The body weights of SAMP8 mice were not significantly different from those of SAMR1 control mice at 3 months of age, but at 10 months of age, their weight gain was significantly less than that of SAMR1 (Table 1). Compared to SAMR1, SAMP8 presented a lower food intake, but food intake was enhanced in the group fed a diet containing HHQ. Although food intake decreased with aging, the HHQ-fed group maintained a high food intake comparable to that of SAMR1. Compared to SAMR1, SAMP8 mice presented a significantly reduced weight gain at 10 months of age.

HHQ and CPP intakes were determined from the mean body weight and food intake. SAMP8 mice ingested 0.03 mg HHQ and 3 mg CPP per gram of body weight (g) at a younger age (Table 2). These concentrations were considerably higher than those commonly consumed in coffee. At an older age, the intake of HHQ and CPP decreased slightly as food intake declined.

Two mice from the HHQ group showed symptoms of polyuria and glucosuria from about 4 months of age (Table 3). The mice fed HHQ for about 2 months indicated the mutation. One of the two mice died at 7 months of age, while the other survived to 10 months of age. Mice that exhibited glycosuria did not have a high mortality rate. However, the 10-month-old survival rate for the HHQ-reared group was 58.3% (7 of 12 mice survived), which tended to be lower than the control SAMP8 mice (9 of 12 mice survived). Renal glucosuria is a disease caused by mutations in sodium–glucose cotransporter 2 (SGLT2), which is specifically expressed in the kidney. No animals in the other groups showed symptoms of polyuria or glucosuria. In addition, SAMP8 mice have never been previously reported to exhibit polyuria or urinary sugar. Therefore, the ingestion of HHQ among SAMP8 mice could be an important factor underlying the development of glucosuria.

### 2.2. Measurement of Hydrogen Peroxide in Serum

Since HHQ was reported to generate relatively large amounts of reactive oxygen species, the amount of hydrogen peroxide in serum was measured. The results showed that SAMP8 mice had higher levels of H_2_O_2_ in serum than SAMR1 mice at the young age of 3 months, with higher levels observed in the HHQ-fed group (Figure 2). On the other hand, H_2_O_2_ levels were significantly lower in younger mice fed both HHQ and CPP compared to the levels in those fed HHQ. At an older age, the amount of H_2_O_2_ was higher than that in younger mice and did not differ between groups. The level also increased in SAMR1 mice with aging.

### 2.3. Measurement of 8-Oxoguanine DNA Glycosylase 1 (Ogg1) in the Liver, Kidney, and Hippocampus

The expression of Ogg1, which repairs the oxidative modifications of DNA, was measured in SAMP8 and SAMR1 mice. We compared the results in the liver and hippocampus between the younger and older groups. The kidneys of young mice were not sampled, so comparisons were made between groups of old mice. The expression level in SAMP8 mice of the younger control group was taken as the reference value of 1. Although the results are presented separately for the younger and older groups, all data were compared.

In the liver, the expression of Ogg1 in the control SAMP8 mice was similar to that in the SAMR1 mice (Figure 3). The expression was lower in the SAMP8 mice that ingested CPP compared to that in SAMR1 mice, although there were no significant differences between the groups of SAMP8 mice. There seemed to be little effect of HHQ or CPP intake on the SAMP8 livers. On the other hand, the expression levels in the hippocampus of young SAMP8 mice fed HHQ and/or CPP were significantly lower than those in SAMR1. In particular, the lowest values were observed in the HHQ-fed group. Since Ogg1 plays an important role in the repair of 8-oxoguanie added via oxidative DNA damage, the reduced expression of Ogg1 in the younger hippocampus due to CPP and HHQ intake may have implications for brain function. No significant differences were found between the older groups. In the kidney, the expression of *Ogg1* was significantly lower in SAMP8 mice than in SAMR1 mice at 10 months, while the expression in the SAMP8 group fed both CPP and HHQ was as high as that of SAMR1 mice.

### 2.4. Measurement of Akt, Nrf2, and HO-1 in the Liver and Kidney

We investigated the type of oxidative injury to the liver and kidneys caused by H_2_O_2_ in the serum, which was elevated via HHQ ingestion. The mRNA expression of *Akt* and *Nrf2*, which are activated by oxidative stress, was also examined. The expression level in SAMP8 mice from the younger control group was shown as the reference value of 1. *Akt* expression was significantly elevated in the livers of mice fed both HHQ and CPP at 3 months of age (Figure 4). At 10 months of age, *Akt* expression was significantly elevated in the livers of mice that had ingested CPP or both HHQ and CPP.

On the other hand, the expression of *Nrf2* was not yet elevated in the liver at 3 months of age in the HHQ-fed group, but the expression was approximately 2-fold higher at 10 months of age compared to that at 3 months of age in SAMP8 mice.

The activation or increased expression of Nrf2 induces the expression of antioxidant enzymes such as HO-1. A comparison of the changes in HO-1 expression showed that the elevated expression in aged SAMP8 mice was significantly suppressed in the livers of the HHQ and CPP co-fed groups. Serum H_2_O_2_ levels were significantly lower in the CPP and HHQ coadministration group, suggesting that oxidative stress was reduced in the mice fed HHQ and CPP. The system of Akt-Nrf2-HO-1 was not significantly altered by HHQ intake. Serum H_2_O_2_, which was elevated in young mice with HHQ ingestion, did not enhance HO-1 expression in young livers. However, this activity was enhanced in the liver and kidney of old mice compared to that in SAMR1 mice.

### 2.5. Effects of HHQ and CPP Intake on Cognitive Function

There was an increase in search time for novel objects in the CPP intake group compared to that in the control group and a trend toward a decrease in the HHQ intake group, but no statistically significant differences were found between these groups (Figure 5).

## 3. Discussion

The effects of HHQ and CPP were examined in SAMP8 mice. The results showed that two out of twelve (16.7%) mice fed HHQ showed symptoms of glucosuria. SAMP8 mice presented reduced activity of Ogg1, one of the enzymes that remove 8-oxoguanine, which accumulates due to oxidative damage to DNA [9]. In SAMP8 mice, the low activity of Ogg1 is thought to be involved in the acceleration of aging and a decline in brain function [10,14]. Ogg1 plays major role in the repair of the oxidatively damaged DNA [15]. As an index of oxidative stress, we measured the serum H_2_O_2_ concentrations of mice fed HHQ and CPP. The results showed that the serum H_2_O_2_ concentration was elevated in younger SAMP8 mice compared to that in SAMR1, which exhibited normal aging. Additionally, the H_2_O_2_ concentration was significantly increased in the HHQ intake group but suppressed when HHQ and CPP were combined.

HHQ was reported to produce ROS and cause DNA single-strand breaks [4]. The symptoms of polyuria and glucosuria suggested that the mutation in sodium–glucose cotransporter 2 (SGLT2) was caused by the ingestion of HHQ in SAMP8 mice with a low activity of Ogg1. Elevated hydrogen peroxide in the blood may also damage organs other than the kidneys, but mutations in SGLT2 in the kidneys can be easily detected as polyuria and glucosuria, which may have led to this discovery. In addition, SGLT2 may be one of the targets of increased H_2_O_2_, as H_2_O_2_ was reported to inhibit SGLT2 and Na/K-ATPase in experiments using cultured mouse proximal tubular cells [16]. SGLT2 is predominantly expressed in the anterior half of the proximal tubule, and approximately 90% of the sugar in the primary urine is reabsorbed by SGLT2. The remaining 10% is transported by another sodium–glucose cotransporter, SGLT1 [17]. Despite the fact that SGLT2 plays a major role in glucose reabsorption, hypoglycemia is known to be less likely to occur in patients with familial glycosuria [18]. Consequently, mutations in SGLT2 are not serious enough to cause lethal injury. In the SAMP10 line, which, like SAMP8, exhibits accelerated aging, a strain of mice that exhibits SGLT2 mutations, SAMP10-ΔSglt2, was previously established [19]. In this case, the mutation was caused by spontaneous mutation, suggesting that the SAMP8 mouse, which has a similar genetic background, is also a mutation-prone strain. The use of the mutagen N-ethyl-N-nitrosourea in creating SGLT2 mutant mice [20] indicates that HHQ is also a potential mutagen for SAMP8 mice.

HHQ is efficiently generated from free or chlorogenic-acid-bound quinic acid moieties, and in addition from carbohydrates and amino acids [21]. HHQ is the quantitatively predominant hydrobenzene in coffee extracts, with a concentration of 32.5 mg/L in freshly prepared very deep roasted coffee samples (5.4 g coffee/100 mL water) and 46.4 mg/L for very light roasted coffee, respectively [22]. However, HHQ in coffee is used as one of the important precursors of aroma binding and its amount decreases rapidly [22]. It has been reported that reducing the amount of HHQ in coffee promotes the postprandial fat oxidation effect of CPP [23]. Furthermore, the effects of HHQ in rat thymocytes and thymic lymphocytes have been investigated [24,25]. Lower HHQ concentrations may decrease the risk of oxidative injury.

The amount of polyphenols in a cup of coffee totals about 0.3 g. In total polyphenols, the CPP content decreases to 47–3% as the degree of roasting increases [26]. It was also reported that the average is about 7.2% in instant coffee [27]. Since the amount of CPP used here is higher than that generally consumed in coffee, we hypothesize that CCP might act as a pro-oxidant like HHQ. For example, epigallocatechin gallate (EGCG), the primary polyphenolic ingredient in green tea, can act either antioxidatively or prooxidatively depending on the dose and its biological environment [28]. CPP has less antioxidant activity than EGCG [29], but it was reported that CPP selectively modulates EGCG-derived reactive oxygen species and that the simultaneous intake of EGCG and CPP decreases H_2_O_2_ production [30]. Similarly, coadministration of CPP with HHQ significantly reduced the amount of H_2_O_2_ in serum, suggesting that CPP acts as an antioxidant and reduces oxidative stress when coadministered with HHQ. When HHQ and CPP coexisted, CPP acted as an antioxidant to counteract the effects of HHQ.

It was previously reported that mutations in *Ogg1* (p.R304W) reduce enzyme activity regardless of organ or age, but do not alter mRNA or protein levels [9]. However, *Ogg1* expression was most significantly decreased in the hippocampus of the HHQ ingestion group. Since it was suggested that reduced *Ogg1* expression may affect brain function [14], we applied a behavioral cognition test. No statistically significant changes were observed with HHQ or CPP ingestion. The expression of REST and IL-1β in the hippocampus was compared between groups because of their association with brain function and inflammatory responses [31]. These findings suggest that HHQ ingestion has no significant effect on brain function.

On the other hand, in older kidneys, the simultaneous ingestion of HHQ and CPP significantly increased *Ogg1* expression compared to the ingestion of HHQ or CPP alone. This result may be related to the lack of glucosuria in the HHQ and CPP coadministration group. In addition, it was suggested that CPP could counteract the effects of HHQ.

The human body can neutralize the harmful effects of oxidative damage through the production of endogenous antioxidants (superoxide dismutase, catalase, glutathione, melatonin). In addition, vitamins A, B, C, E, coenzyme Q-10, selenium, flavonoids, lipoic acid, carotenoids, and lycopene in foods have antioxidant capacity [32]. Polyphenols presented in the fruit wines have been reported to show the ability to activate enzymes of oxidative stress in isolated synaptosomes during experimentally induced oxidative stress by hydrogen peroxide [33]. Therefore, it is quite possible that HHQ taken into the body is counteracted in its action not only by CPP but also by endogenous antioxidants and antioxidants contained in food. These effects need to be taken into account in future studies.

## 4. Materials and Methods

### 4.1. Animals and Reagents

SAMP8 mice were used as the experimental animals. Four-week-old male mice were purchased from Japan SLC, and three mice per cage were kept under a 12 h light/dark cycle (8:00–20:00 light period) at room temperature under a temperature of 23 ± 1 °C and humidity of 55 ± 5%. Acclimation rearing was conducted until the animals reached 2 months of age. During the acclimation period, all mice were fed AIN-93G (Oriental Yeast Co., Ltd., Tokyo, Japan), which was used as a control. Feed and drinking water were provided ad libitum, and HHQ and CPP were administered to the mice as a solid diet, with AIN-93G (CD) as the base feed. HHQ (1,2,4-benzenetriol) was purchased from the Fujifilm Wako Pure Chemical Corporation (Osaka, Japan).

Animal care and experiments were conducted in accordance with the University of Shizuoka’s guidelines for animal experiments. For SAMP8 mice, the number of mice per group was 18, and 4 groups were established. Mice were divided into a control group (CD), a group that received food containing 0.02% HHQ (HHQ), a group that received food containing 2% CPP (CPP), and a group that received food containing both HHQ and CPP (HHQ + CPP). SAMR1 mice that presented normal aging were given CD as a reference group.

After a month of acclimation, 6 mice per group were dissected after 1 month of feeding (3 months of age) for SAMP8 and SAMR1 mice to extract samples from the young mice. Then, the experiment continued, and the surviving mice were dissected after 8 months of feeding (10 months of age) to extract samples from old mice. Blood, liver, and hippocampus samples were collected to be used as samples. In addition, kidneys were collected to obtain samples from only old mice.

The amount of food intake, body weight gain, and survival time were measured during this period. Before dissection, we conducted a novel object recognition test to examine the effects on brain function. All experimental protocols were approved by the University of Shizuoka’s Laboratory Animal Care Advisory Committee (approval No. 225354, 3 March 2022) and performed in accordance with the guidelines of the US National Institutes of Health for the care and use of laboratory animals.

### 4.2. Preparation of CPP

CPPs were extracted from green coffee beans with hot water and spray-dried according to the previous report [34]. The CPP composition was analyzed via high-performance liquid chromatography by using a Cadenza CD C18 column (4.6 mm i.d. × 150 mm; Intact, Kyoto, Japan), a Prominence Inert LC system (Shimadzu, Kyoto, Japan), and with a UV detector (325 nm). The compound 5-caffeoylquinic acid (CQA, Sigma-Aldrich, Poole, UK) was used to generate a calibration curve. The total polyphenol content was 42.9% and composed of various polyphenols as follows: 12.5% 3-CQA, 11.7% 4-CQA, 14.0% 5-CQA, 1.5% 3-feruloyquinic acid (FQA), 1.5% 4-FQA, and 1.8% 5-FQA. The CPP preparation contained no caffeine.

### 4.3. Assessment of Urinary Glucose

Diagnosis chips were used to assess glucosuria (Pretest 5bII, Wako, Wako Pure Chemicals Industries, Ltd., Osaka, Japan).

### 4.4. Measurement of Hydrogen Peroxide in Serum

Since HHQ was reported to generate relatively large amounts of reactive oxygen species, the amount of hydrogen peroxide in the serum was measured. Blood collected from the abdominal aorta was centrifuged using Capiject (CJ-AS, Terumo Co., Tokyo, Japan). The obtained serum was stored at −80 °C until use. Hydrogen peroxide was measured using a commercially available kit (OxiSelect™ Hydrogen Peroxide/Peroxidase Assay Kit (Colorimetric) STA-84, Cell Biolabs. Inc., San Diego, CA, USA). The procedure was followed as described, and absorbance was measured at 540 nm.

### 4.5. Novel Object Recognition Test

This test was performed to evaluate long-term memory performance. In order to familiarize the mice with the gray acrylic apparatus (50 cm long × 50 cm wide × 40 cm high), each mouse was allowed to explore the apparatus for 5 min. The next day, two identical objects (triangular pyramids) were placed in the apparatus as a training test, and the contact time with each object was recorded for 5 min. A day later, one of the objects was replaced with a new object (quadrangular prism), and the contact time with each object was recorded for 5 min. In the memory retention trial, the ratio of search time for the new object to the total search time was calculated and used as an index of cognitive function. Higher values indicate a higher memory retention capacity.

### 4.6. Quantitative Real-Time Reverse Transcription PCR (qRT-PCR)

The mice were anesthetized with isoflurane. The brains were carefully dissected, and the hippocampus and cerebral cortex were immediately frozen. Real-time PCR was performed on the brain samples to compare the expression changes in each gene. Total RNA was extracted from tissues using a purification kit (NucleoSpin^®^ RNA, 740955, Takara Bio Inc., Shiga, Japan) in accordance with the manufacturer’s protocol. The obtained RNA was converted to cDNA using a PrimeScript™ RT Master Mix kit (RR036A, Takara Bio Inc.). A real-time quantitative PCR analysis was performed using the PowerUp™ SYBR™ Green Master Mix (A25742, Applied Biosystems Japan Ltd., Tokyo, Japan) and automated sequence detection systems (StepOne, Applied Biosystems Japan Ltd.). Relative gene expression was measured using previously validated primers for the *Ogg1* [35], *Akt* [36], *Nrf2* [37], *HO-1* [38], *Rest* [39], and *IL-1β* [40] genes. The primer sequences are set out in Table 4. cDNA derived from transcripts encoding β-actin was used as the internal control.

### 4.7. Statistical Analyses

The results are expressed as the mean ± SEM (hydrogen peroxide levels in Figure 2, mRNA expression levels in Figure 3 and Figure 4, and the novel object recognition (%) in Figure 5). A statistical analysis was performed using one-way ANOVA, and statistical significance was set at *p* < 0.05. Confidence intervals and significance of differences in means were estimated using the Tukey–Kramer significant difference method.

## 5. Conclusions

HHQ, a byproduct in roasted coffee, can cause DNA mutations in SAMP8 mice, which have a mutation in the *Ogg1* gene and are sensitive to oxidative stress. HHQ was shown to be able to mutate against SAMP8, e.g., producing glucosuria. The results indicate that HHQ enhances oxidative injury to oxidative-damage-sensitive SAMP8 mice. When HHQ and CPP coexisted in the proportions normally found in roasted coffee, they were shown to decrease the amount of hydrogen peroxide in the blood. This result suggests that CPP acts in an antioxidant manner to counteract the effects of HHQ. Thus, in roasted coffee, it is likely that the action of HHQ is suppressed by CPP. However, HHQ-reduced coffee may be preferable in order to avoid potential adverse effects.

## Figures and Tables

**Figure 1 ijms-25-00720-f001:**
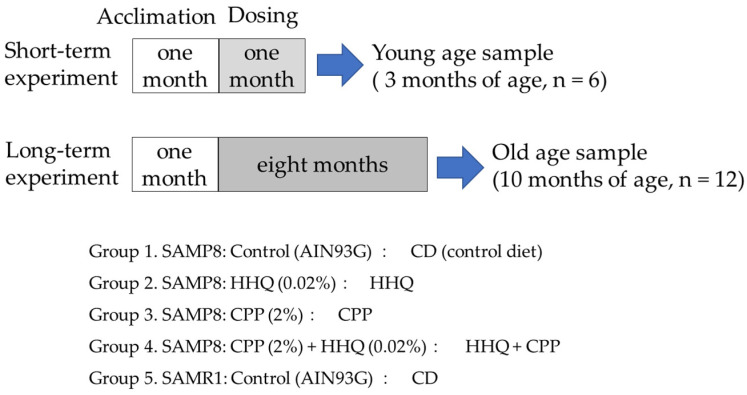
Experimental protocol.

**Figure 2 ijms-25-00720-f002:**
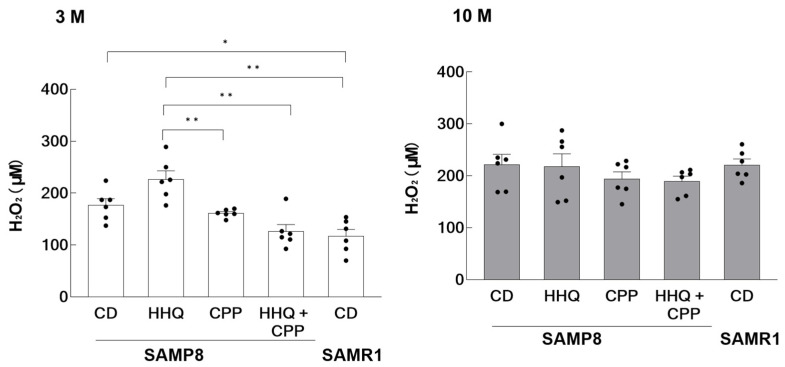
Hydrogen peroxide level in the serum of younger (3 M) and older (10 M) mice. Each column bar represents the mean ± SEM (*n* = 6) overlaid on scatter plots (* *p* < 0.05 and ** *p* < 0.01, Tukey’s honestly significant difference method). Black dots indicate the values for each mouse.

**Figure 3 ijms-25-00720-f003:**
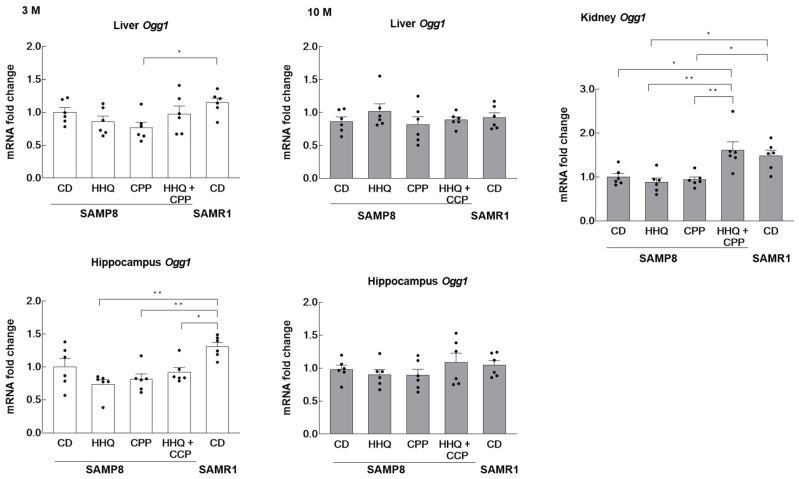
Effects of HHQ and CPP intake on *Ogg1* mRNA levels in the liver, hippocampus, and kidney of younger (3 M) and older (10 M) mice. For kidneys, comparisons were made only in old mice. Each column bar represents the mean ± SEM (*n* = 6) overlaid on scatter plots (* *p* < 0.05 and ** *p* < 0.01, Tukey’s honestly significant difference method). Black dots indicate the values for each mouse.

**Figure 4 ijms-25-00720-f004:**
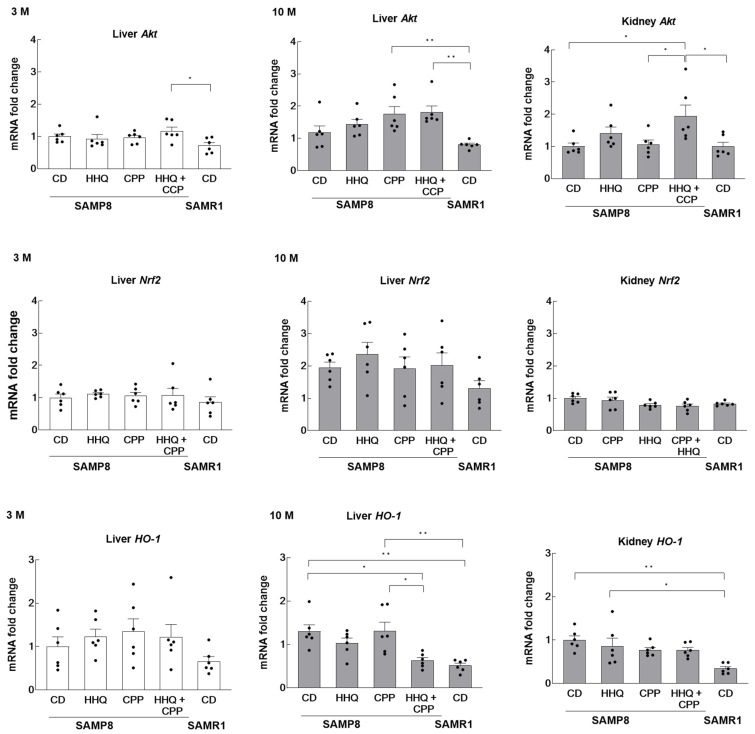
The levels of *Akt, Nrf2,* and *HO-1* in the liver and hippocampus of younger (3 M) and older (10 M) mice. For kidneys, comparisons were made only in older mice. Each column bar represents the mean ± SEM (*n* = 6) overlaid on scatter plots (* *p* < 0.05 and ** *p* < 0.01, Tukey’s honestly significant difference method). Black dots indicate the values for each mouse.

**Figure 5 ijms-25-00720-f005:**
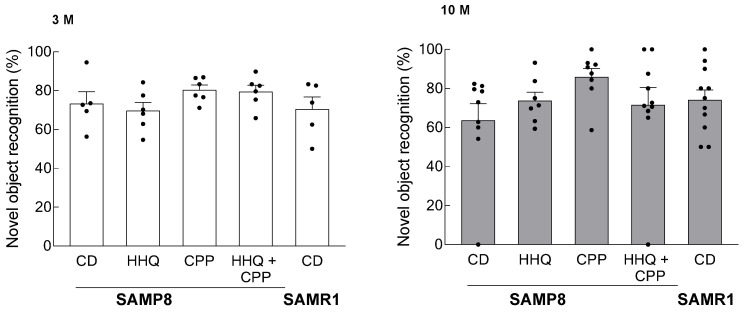
Novel object recognition test in younger (3 M) and older (10 M) mice. Each column bar represents the mean ± SEM (*n* = 6) overlaid on scatter plots. Black dots indicate the values for each mouse.

**Table 1 ijms-25-00720-t001:** Effects of HHQ and CPP intake on body weight and food intake.

Mouse	Group	Body Weight (g)		Food Intake (g/Day)	
		3 M	10 M	3 M	10 M
SAMP8	Control	29.6 ± 1.22	b 30.8 ± 1.67	c 4.24 ± 0.11	c 3.85 ± 0.06
	HHQ	31.0 ± 1.30	b 32.1 ± 1.78	ab 4.62 ± 0.08	ab 4.48 ± 0.07
	CPP	27.6 ± 1.49	b 27.2 ± 0.69	c 4.39 ± 0.08	c 3.94 ± 0.07
	HHQ + CPP	27.0 ± 0.87	b 28.1 ± 1.24	abc 4.54 ± 0.06	c 4.05 ± 0.08
SAMR1	Control	30.3 ± 1.68	a 41.9 ± 1.75	ab 4.74 ± 0.10	ab 4.42 ± 0.05

The difference between means is statistically significant (*p* < 0.05) for groups with different alphabetical symbols.

**Table 2 ijms-25-00720-t002:** HHQ and CPP intake.

Group	Intake/Day	3 M		10 M	
		HHQ	CPP	HHQ	CPP
HHQ	mg/mouse	0.92 ± 0.02		0.90 ± 0.01	
	mg/g BW	0.03 ± 0.00		0.03 ± 0.00	
CPP	mg/mouse		87.7 ± 1.63		78.9 ± 1.40
	mg/g BW		b 3.01 ± 0.10		b 2.48 ± 0.05
HHQ + CPP	mg/mouse	0.91 ± 0.01	90.8 ± 1.19	0.81 ± 0.02	80.9 ± 1.53
	mg/g BW	0.03 ± 0.00	a 3.29 ± 0.07	0.03 ± 0.00	a 2.67 ± 0.06

BW; body weight. The difference between means is statistically significant (*p* < 0.05) for groups with different alphabetical symbols.

**Table 3 ijms-25-00720-t003:** Effect of HHQ intake on glucosuria and survival.

HHQ Group	Mouse (n) (%)	Dead Mice (n) (Age)	Survives to 10 M (n) (%)
Glucosuria	2 (16.6%)	1 (7.0 M)	1 (50%)
Normal urine	10 (83.3%)	4 (2.4, 7.6, 8.1 and 9.9 M)	6 (60%)

**Table 4 ijms-25-00720-t004:** Sequence of primers used in the qRT-PCR.

Gene	Forward Sequence	Reverse Sequence	Ref.
*Ogg1*	GCCAACAAAGAACTGGGAAA	CCCTCTGGCCTCTTAGATCC	[35]
*Akt*	ATCCCCTCAACAACTTCTCAGT	CTTCCGTCCACTCTTCTCTTTC	[36]
*Nrf2*	GTCTTCACTGCCCCTCATC	TCGGGAATGGAAAATAGCTCC	[37]
*HO-1*	TGCAGGTGATGCTGACAGAGG	GGGATGAGCTAGTGCTGATCTGG	[38]
*Rest*	ATCGGACGCGGGTAGCGAG	GGCTGCCAGTTCAGCTTTCG	[39]
*IL-1β*	GCAACTGTTCCTGAACTCAACT	ATCTTTTGGGGTCCGTCAACT	[40]
*β-actin*	TGACAGGATGCAGAAGGAGA	GCTGGAAGGTGGACAGTGAG	

## Data Availability

The data presented in this study are available upon request from the corresponding author.

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
