# Peer review of "DNA Mutagenicity of Hydroxyhydroquinone in Roasted Coffee Products and Its Suppression by Chlorogenic Acid, a Coffee Polyphenol, in Oxidative-Damage-Sensitive SAMP8 Mice"

_ijms, 2024, doi:10.3390/ijms25020720_

Round 1

Reviewer 1 Report

Comments and Suggestions for Authors

Dear Authors,

This study offers information regarding biological activity of polyphenols from coffee which was observed in animal model.

In the section materials and methods subsection 2.1. is highlighted composition of CPP used in the study. The caracterization of the CPP is very important and paragraph which is related to this data have to be new subsection at the begginig of the section results. It is not possible to observe biological activity of natural product if you do not have chemical composition of the product.

The reference related to the HPLC method used for determination of CPP composition is missing.

What is the type of HPLC system used in analysis? The name of company which produced system is missing? The name of city and country from which is company is missing too.

Which column was used for separation?

Which detector was used?

Did you used external standards for determination?

What were conditions of determination of polyphenolic compounds from CPP?

A lot of important information related to the qualitative and quantitative analysis missing. Insert table in which will be data (ionization mode ESI, MRM transition, cone voltage, collision energy and retention time) related to quantification of polyphenols from CPP. It is very important. Insert this table in manuscript.  

In the line 269 it is indicated that total polyphenol content was 42,9%. How did you obtain it result? Which method was used for determination?

In the subsection 4.6. highlight which parameters were used in the statistical tests applied for analysis of data presented in this manuscript

In the line 237 highlgiht that polyphenols presented in the fruit wines showed ability to activate enzymes of oxidative stress in isolated synaptosomes during experimentally induced oxidaive stress by hydrogen peroxide. Kindly consider to cite Agronomy, 11(7), (2021) 1414.

Wish you all the best in the future work,

Author Response

Reviewer 1

Dear Authors,

This study offers information regarding biological activity of polyphenols from coffee which was observed in animal model.

In the section materials and methods subsection 2.1. is highlighted composition of CPP used in the study. The caracterization of the CPP is very important and paragraph which is related to this data have to be new subsection at the begginig of the section results. It is not possible to observe biological activity of natural product if you do not have chemical composition of the product.

Thank you for your peer review. We have followed your suggestion and designated the preparation of CPP as Section 4.2.

The reference related to the HPLC method used for determination of CPP composition is missing.

What is the type of HPLC system used in analysis? The name of company which produced system is missing? The name of city and country from which is company is missing too.

Which column was used for separation?

Which detector was used?

Did you used external standards for determination?

What were conditions of determination of polyphenolic compounds from CPP?

The above items are described in Section 4.2 as follows: The CPP composition was analyzed via high-performance liquid chromatography by using a Cadenza CD C18 column(4.6 mm i.d. × 150 mm; Intact, Kyoto, Japan), a Prominence Inert LC system (Shimadzu, Kyoto, Japan) and with a UV detector (325 nm). 5-caffeoylquinic acid (CQA, Sigma-Aldrich, Poole, UK) was used to generate a calibration curve. The total polyphenol content was 42.9% and composed of various polyphenols as follows: 12.5% 3-CQA, 11.7% 4-CQA, 14.0% 5-CQA, 1.5% 3-feruloyquinic acid (FQA), 1.5% 4-FQA, and 1.8% 5- FQA. The CPP preparation contained no caffeine.

A lot of important information related to the qualitative and quantitative analysis missing. Insert table in which will be data (ionization mode ESI, MRM transition, cone voltage, collision energy and retention time) related to quantification of polyphenols from CPP. It is very important. Insert this table in manuscript.  

Since the analysis of CPPs was performed according to the method already reported so far (Ishida K et al, 2020), it was not included in the new results (Section 2) in this paper. We showed its citation in Section 4.2.

In the line 269 it is indicated that total polyphenol content was 42,9%. How did you obtain it result? Which method was used for determination?

As described in Section 4.2, 5-caffeoylquinic acid (CQA, Sigma-Aldrich, Poole, UK) was used to generate a calibration curve and CPP composition was analyzed by HPLC..

In the subsection 4.6. highlight which parameters were used in the statistical tests applied for analysis of data presented in this manuscript.

We revised it.

In the line 237 highlgiht that polyphenols presented in the fruit wines showed ability to activate enzymes of oxidative stress in isolated synaptosomes during experimentally induced oxidaive stress by hydrogen peroxide. Kindly consider to cite Agronomy, 11(7), (2021) 1414.

Wish you all the best in the future work,

Thank you for your suggestion.

We have cited it in our discussion.

Reviewer 2 Report

Comments and Suggestions for Authors

Notes and comments to the authors:

1. The topic of the manuscript is particularly relevant, as it is related to biochemical genetics and more precisely studies DNA Mutagenicity of Hydroxyhydroquinone in Roasted Coffee Products and Its Suppression by Chlorogenic Acid, a Coffee Polyphenol, in Oxidative-Damage-Sensitive SAMP8 Mice;

2. In the Introduction section, the authors have made a detailed analysis of research on the influence of polyphenols in roasted coffee beans on oxidative stress at the cellular and organismal level. The anti-aging effect of a number of antioxidants as a means of neutralizing pro-oxidants has also been examined;

3. The results are presented through relatively well-prepared figures and tables. However, Figure 4 is quite cluttered and contains 6 components. On the other hand, this is a convenience for easier interpretation of the obtained results, since the changes in the liver and hippocampus of younger and older mice are compared;

4. The Discussion section best demonstrates the erudition of the authors and their ability to dig deep into the bowels of biochemical genetics and gene expression. In my opinion, the authors have conducted a valuable discussion analyzing the problem in depth, revealing the intimate mechanisms of the occurrence of oxidative stress, the expression of specific genes related to antioxidant protection and concerning the anti-aging effect of specific components of coffee;

5. In the Material and methods section, the biological models /SAMP8 mice were used as the experimental animals/ are discussed in detail. The methods used are modern, modern and provide real and reliable results. Particularly impressive is the Quantitative Real-Time Reverse Transcription PCR (qRT-PCR) analysis, which is largely innovative for this kind of research;

6. A valuable conclusion was made that CPP acts in an antioxidant manner to counteract the effects of HHQ. Thus, in roasted coffee, it is likely that the action of HHQ is suppressed by CPP.

Negative findings and recommendations to the authors:

1. Whether the requirements for humane treatment of experimental animals are not clearly observed. It is only stated that "animal care and experiments were conducted in accordance with the University of Shizuoka's guidelines for animal experiments".

2. The technological influence of the coffee roasting temperature on the production of hydroxyhydroquinone, as a powerful pro-oxidant, has not been discussed. In addition, there are other factors supporting the oxidative balance in the body such as some vitamins /C, E, Lipoic acid, Glutathione, etc./ as well as minerals (selenium, zinc, etc.).

3. In the Conclusions section, a recommendation to medicine and dietetics about the importance and benefits of coffee could be included, which would be an important scientific and applied contribution.

Comments on the Quality of English Language

Notes on the English language of the manuscript.

The article is written in competent English. However, I recommend its final polishing by an English-speaking editor.

Author Response

Reviewer 2

Notes and comments to the authors:

  1. The topic of the manuscript is particularly relevant, as it is related to biochemical genetics and more precisely studies DNA Mutagenicity of Hydroxyhydroquinone in Roasted Coffee Products and Its Suppression by Chlorogenic Acid, a Coffee Polyphenol, in Oxidative-Damage-Sensitive SAMP8 Mice;
  2. In the Introduction section, the authors have made a detailed analysis of research on the influence of polyphenols in roasted coffee beans on oxidative stress at the cellular and organismal level. The anti-aging effect of a number of antioxidants as a means of neutralizing pro-oxidants has also been examined;
  3. The results are presented through relatively well-prepared figures and tables. However, Figure 4 is quite cluttered and contains 6 components. On the other hand, this is a convenience for easier interpretation of the obtained results, since the changes in the liver and hippocampus of younger and older mice are compared;
  4. The Discussion section best demonstrates the erudition of the authors and their ability to dig deep into the bowels of biochemical genetics and gene expression. In my opinion, the authors have conducted a valuable discussion analyzing the problem in depth, revealing the intimate mechanisms of the occurrence of oxidative stress, the expression of specific genes related to antioxidant protection and concerning the anti-aging effect of specific components of coffee;
  5. In the Material and methods section, the biological models /SAMP8 mice were used as the experimental animals/ are discussed in detail. The methods used are modern, modern and provide real and reliable results. Particularly impressive is the Quantitative Real-Time Reverse Transcription PCR (qRT-PCR) analysis, which is largely innovative for this kind of research;
  6. A valuable conclusion was made that CPP acts in an antioxidant manner to counteract the effects of HHQ. Thus, in roasted coffee, it is likely that the action of HHQ is suppressed by CPP.

 Thank you so much for reviewing our manuscript. The authors deeply appreciate the reviewers' comments.

Negative findings and recommendations to the authors:

  1. Whether the requirements for humane treatment of experimental animals are not clearly observed. It is only stated that "animal care and experiments were conducted in accordance with the University of Shizuoka's guidelines for animal experiments".

We have added a note that all experimental protocols were performed in accordance with the guidelines of the US National Institutes of Health for the care and use of laboratory animals.

  1. The technological influence of the coffee roasting temperature on the production of hydroxyhydroquinone, as a powerful pro-oxidant, has not been discussed. In addition, there are other factors supporting the oxidative balance in the body such as some vitamins /C, E, Lipoic acid, Glutathione, etc./ as well as minerals (selenium, zinc, etc.).

We added them in the discussion.

  1. In the Conclusions section, a recommendation to medicine and dietetics about the importance and benefits of coffee could be included, which would be an important scientific and applied contribution.

We added our recommendation.

Reviewer 3 Report

Comments and Suggestions for Authors

The paper entitled „DNA Mutagenicity of Hydroxyhydroquinone in Roasted Coffee Products and Its Suppression by Chlorogenic Acid, a Coffee Polyphenol, in Oxidative-Damage-Sensitive SAMP8 Mice“ by Unno et al. is very well written, so I recommend to be published in IJMS after a minor revision of the manuscript.

1.

According to Instructions for Authors, acronyms/abbreviations/initialisms should be defined the first time they appear in each of three sections: the abstract; the main text; and the first figure or table. When defined for the first time, the acronym/abbreviation/initialism should be added in parentheses after the written-out form. I recommend the authors revise the abstract and main text according to these instructions (e.g.: SAMR in line 22; CD in Figure 1, etc).

2.

In Figure 1, what is marked by the ordinal numbers 1 to 5? I assume it is the treatments, so write that on the figure.

3.

The extraction of CPP from coffee (line 267) has been poorly described, so it remains unknown what amount of coffee and in what form it was used for extraction, as well as the ratio of coffee to water. How long did the extraction take? Please describe the extraction procedure in more detail!

4.

How did you determine sample sizes in laboratory animal experiments? By what method?

5.

The results of the HPLC analysis of CPP should be reported in Chapter 2, not Chapter 4!

Author Response

Reviewer 3

The paper entitled “DNA Mutagenicity of Hydroxyhydroquinone in Roasted Coffee Products and Its Suppression by Chlorogenic Acid, a Coffee Polyphenol, in Oxidative-Damage-Sensitive SAMP8 Mice“ by Unno et al. is very well written, so I recommend to be published in IJMS after a minor revision of the manuscript.

  1. According to Instructions for Authors, acronyms/abbreviations/initialisms should be defined the first time they appear in each of three sections: the abstract; the main text; and the first figure or table. When defined for the first time, the acronym/abbreviation/initialism should be added in parentheses after the written-out form. I recommend the authors revise the abstract and main text according to these instructions (e.g.: SAMR in line 22; CD in Figure 1, etc).

Thank you so much for reviewing our manuscript.

We revised them.

  1. In Figure 1, what is marked by the ordinal numbers 1 to 5? I assume it is the treatments, so write that on the figure.

We revised it.

  1. The extraction of CPP from coffee (line 267) has been poorly described, so it remains unknown what amount of coffee and in what form it was used for extraction, as well as the ratio of coffee to water. How long did the extraction take? Please describe the extraction procedure in more detail!

CPPs were extracted from green coffee beans with hot water and spray dried, which is similar to the previous reports (https://pubmed.ncbi.nlm.nih.gov/31121203/). The preparation of CPPs is described as Section 4.2.

  1. How did you determine sample sizes in laboratory animal experiments? By what method?

The sample size of the animal experiments was based on our previous animal experiments on CPP (Unno et al, 2022).

  1. The results of the HPLC analysis of CPP should be reported in Chapter 2, not Chapter 4!

Since the analysis of CPPs was performed according to the method already reported so far (Ishida K et al, 2020), it was not included in the new results (Section 2) in this paper. We showed its citation in Section 4.2.

Round 2

Reviewer 1 Report

Comments and Suggestions for Authors

Dear Authors, 

Thank you very much for revised version of manuscript and answers on questions, it is fine for me. 

Best regards,